# Exposure and Health Risk Assessment of Aflatoxin M_1_ in Raw Milk and Cottage Cheese in Adults in Ethiopia

**DOI:** 10.3390/foods12040817

**Published:** 2023-02-14

**Authors:** Haftom Zebib, Dawit Abate, Ashagrie Zewdu Woldegiorgis

**Affiliations:** 1Center for Food Science and Nutrition, Addis Ababa University, Addis Ababa P.O. Box 1176, Ethiopia; 2Livestock and Fishery Core Process, Tigray Agricultural Research Institute, Mekelle P.O. Box 492, Ethiopia; 3Department of Biology, Addis Ababa University, Addis Ababa P.O. Box 1176, Ethiopia

**Keywords:** aflatoxin M_1_, dairy, dietary exposure, risk assessment, cancer risk, public health

## Abstract

Aflatoxin M_1_ (milk toxin) found in milk is formed from the hepatic biotransformation of AFB_1_ (aflatoxin B_1_) and poses a risk to human health when consumed. The risk assessment of AFM_1_ exposure due to milk consumption is a valuable way to assess health risk. The objective of the present work was to determine an exposure and risk assessment of AFM_1_ in raw milk and cheese, and it is the first of its kind in Ethiopia. Determination of AFM_1_ was conducted using an enzyme-linked immunosorbent assay (ELISA). The results indicated that AFM_1_ was positive in all samples of milk products. The risk assessment was determined using margin of exposure (MOE), estimated daily intake (EDI), hazard index (HI), and cancer risk. The mean EDIs for raw milk and cheese consumers were 0.70 and 0.16 ng/kg bw/day, respectively. Our results showed that almost all mean MOE values were <10,000, which suggests a potential health issue. The mean HI values obtained were 3.50 and 0.79 for raw milk and cheese consumers, respectively, which indicates adverse health effects for large consumers of raw milk. For milk and cheese consumers, the mean cancer risk was 1.29 × 10^−6^ and 2.9 × 10^−6^ cases/100,000 person/year, respectively, which indicates a low risk for cancer. Therefore, a risk assessment of AFM_1_ in children should be investigated further as they consume more milk than adults.

## 1. Introduction

Milk contains necessary macro- and micro-nutrients for the growth, development, and maintenance of human health. However, milk safety is a great concern in developing countries where the production of milk takes place under poor production and management practices along the value chain actors [1,2]. Milk contamination occurs due to chemical, biological, and/or physical hazards. Chemical hazard contamination mainly includes aflatoxins and heavy metals or residues from veterinary drugs or pesticides. Among these, aflatoxins are metabolic by-products produced mainly by molds, and *Aspergillus flavus* and *Aspergillus parasiticus* resulting from poor preharvest, postharvest, and storage practices of feed. The major hydroxyl metabolite of aflatoxin M_1_ (AFM_1_), which is formed when lactating cows consume feed containing aflatoxin B_1_ (AFB_1_), is found in milk and thus poses a threat to human health through consumption [3,4].

A risk assessment includes four stages, namely hazard identification, hazard characterization, exposure assessment, and risk characterization [5,6]. A risk characterization is used to compare the predicted daily exposure to toxicological threshold values of non-carcinogenic mycotoxins, such as tolerable daily intake or provisional maximum tolerable daily intake (PMTDI). Furthermore, it assists in determining the amount of public health concern and in setting a priority for risk management actions [7,8].

Human exposure to AFM_1_ mainly occurs through the consumption of contaminated milk. To protect public health, many countries have set maximum permissible limits for AFM_1_ in the milk products through risk assessment studies [5]. However, Ethiopia has adopted international standards [9,10] or established specifications for residue maximum limits in the dairy products. No tailored and dedicated risk assessment that represents the country has been conducted considering the proper contamination of AFM_1_ in milk and milk products and consumption data among dairy products of Ethiopian consumers.

There are data available on the contamination of milk and dairy products with AFM_1_ and risk assessment globally [11,12,13,14]. However, there is a scarcity of studies regarding AFM_1_ occurrence in dairy products in Ethiopia. Thus far, only three have been published [15,16,17]. These investigations reported the presence of AFM_1_ occurrence in milk and cottage cheese in particular district areas. Moreover, no science-based exposure assessment or health risk assessment of AFM_1_ in milk and dairy products has been conducted so far in Ethiopia to address the potential health impact (liver cancer development) for the Ethiopian population.

According to [18], risk analysis is required at the regional or national level in order to fully understand risk exposure and to determine the most important public health targets for policy intervention. Therefore, the scientific information generated from this research could be an important input for the development of evidence-based mitigation strategies in reducing and controlling AFM_1_, aiding policy and regulatory organizations in setting standards in the dairy supply chain at a national level. The aim of the present study was to conduct an exposure and health risk of AFM_1_ in raw milk and cottage cheese, and it is the first of its kind in Ethiopia. Prior to conducting the exposure assessment, the level of AFM_1_ in raw milk and cottage cheese was determined along the value chain from three regions of Ethiopia with high milk production potential to provide input data on the contamination.

## 2. Materials and Methods

### 2.1. Study Area

Based on data from the Central Statistical Agency (CSA), the study regions for sample collection were selected for their high milk production potential [19]. Along the value chain, including dairy farmers, collectors, and retailers, these areas include urban and peri-urban regions of Oromia, SNNP (Southern Nation, Nationalities, and People), and Amhara regions. In the SNNP region, Wollayta, Yirgalem, Hawassa, and Dilla; in the Oromia region, Wolmera, Debrezeyt/Bishoftu, Asela, and Selale; and in the Amhara region, Debrebirhan, Gondar Debremerkos, and Bahirdar.

### 2.2. Sample Size and Sampling

Primary samples were collected through a cross-sectional study. Using a simple random method, producers (at the farm level) and retailers were selected, whereas collectors were selected purposefully. Composite samples per product type were obtained by mixing five primary samples together by taking about 50 g per sample to obtain one representative composite sample along the value chain in each region [20]. Table 1 shows composite sample size of milk products along the value chain.

### 2.3. Determination of AFM_1_ Occurrences in Milk Products in Ethiopia

The quantitative analysis of AFM_1_ in raw milk and cheese was conducted by HELICA AFM_1_ ELISA kit (CAT. NO. 961AFLM01M-96; Helica Biosystems Inc., Santa Ana, CA, USA). The HELICA AFM_1_ assay is a solid phase competitive enzyme immunoassay. Polystyrene micro wells were coated with an antibody that has a high affinity for AFM_1_. The ELISA kit had between 0.005 to 0.1 µg/L AFM_1_ standard solutions. For all milk products, the method of sample preparation, calculations, and result interpretation were performed in accordance with the manufacturer’s manual [21]. To bring the absorbance within the range, additional dilutions were applied to samples that fell outside the highest standard concentration range.

For the method validation, a recovery was performed by spiking the AFM_1_ standard (0.05 g/L) into samples of raw milk and cheese. The percentages of recovery were 95.9% and 89% for raw milk and cottage cheese, respectively. Samples that had the lowest concentrations were used for spike tests. The lowest concentration for raw milk was 0.0031 µg/L, whereas for cottage cheese it was 0.0139 µg/L. Intra-assay precision was achieved by running the same sample on a plate and having a within-assay coefficient of variation (CV) of 4.74% and 3.60% for raw milk and cottage, respectively. Inter-assay precision was also carried out by assaying AFM_1_-free milk and the 0.05 µg/L standard (provided with kits) throughout the study duration and had 11.1% and 5.12% CV, respectively. This ensures the validity of the results of the study [22]. The LOD for ELISA method was 0.002 µg/L. Additional information is available in previously published work [22].

### 2.4. Production Procedure for Local Cottage Cheese

Traditional cottage cheese is lactic acid fresh cheese with lower pH value. To prepare this, first raw milk is fermented under ambient temperature for 2–3 days to make traditional *ergo* (traditional yoghurt). Then it is processed, which differs from soft cheese made in other countries, by churning the *ergo* and removing the cream, then gently heating the butter milk and draining off the whey [23]. This is made without addition of rennet and coagulants, which is practiced in different countries.

### 2.5. Raw Milk and Cottage Cheese Consumption Data

Household consumption data are the major part of a risk assessment as they are used as an input to the exposure assessment. The Ethiopian Public Health Institute’s (EPHI) existing national dairy consumption data collected from Oromia, SNNP, and Amhara regions were used for the exposure and health risk assessment estimations. The raw milk and cheese consumption survey consisted of information related to the identification of the consumers’ age, gender, body weight, and volume consumed. Participants in the survey were women of reproductive age (15–45 years), and men aged between 19 and 45 years [24]. According to CSA data, an individual adult woman’s age in Ethiopia ranged from 15 to 49 years, and an individual adult man’s age ranged from 15 to 59 years [25]. The consumption data were collected based on a 24 h dietary recall. The total survey participants were 16,592, although there were proportionally fewer men [24]. There was no information about the measurement of body weight in men. We assumed 65 kg as the average body weight for men in each region [26,27].

### 2.6. Deterministic Exposure Assessment

To determine a deterministic exposure to AFM_1,_ C_ave_ for each regional location, the mean of AFM_1_ concentration, intake of milk products, and body weight were used [27].

Exposure is calculated as
(1)$$\mathrm{EDI}=\frac{C_{\mathrm{ave}}\times \mathrm{IR}}{\mathrm{bw}}\;$$
where estimated daily intake (EDI) is the average daily dose, C_ave_ is the mean concentration of the toxin in the dairy product (μg AFM_1_ per L), IR is the intake rate by the individual consumer of dairy product, and bw is individual’s bodyweight.

### 2.7. Risk Characterization

The health risk of AFM_1_ was characterized using cancer risk [28], margin of exposure [7], and hazard index [5] approaches. The mean and median values of HI, MOE, and cancer risk were calculated using the mean and median daily intake of AFM_1_ and the mean consumption rate of dairy products in the target populations. In present work, we used mean and median values of AFM_1_ and each of the risk characterization parameters as the AFM_1_ data were not normally distributed.

#### 2.7.1. Cancer Risk

JECFA assessed the cancer risk associated with exposure to 1 ng AFB_1_/kg bw/day in a population of 100,000. The obtained upper bounds were 0.049 additional cancer cases/10^5^ people in HBsAg—populations and 0.562 additional cancer cases/10^5^ in HBsAg+ populations [28]. AFM_1_ was about one-tenth as potent as AFB_1_ on carcinogenicity studies, even in sensitive species such as Fischer rats and Rainbow trout [28]. Therefore, the carcinogenic potency of AFM_1_ was calculated to be 0.0562 additional cancer cases/10^5^ for HBsAg+ populations and 0.0049 additional cancer cases/10^5^ for HBsAg−populations. In this study, an overall pooled prevalence (7.4%) of hepatitis B virus (HBV)-positive populations was adopted according to [29]. For HBsAg-negative populations, 92.6% was obtained by differences (100–7.4%). The prevalence data were obtained through systematic review and meta-analysis conducted on five decades of published studies (1968–2015) from biomedical databases.
(2)$$\begin{array}{cc} \mathrm{Average}\;\mathrm{cancer} & \;\mathrm{potency}\\ & =\left(0.0562\times \mathrm{HBsAg}\;\mathrm{negative}\;\mathrm{population}\;\right)\\ & +(0.0049\times \;\mathrm{HBsAg}\;\mathrm{positive}\;\mathrm{populations}\\ & /\mathrm{prevalence}\;\mathrm{in}\;\mathrm{Ethiopia}) \end{array}\;\mathrm{Average}\;\mathrm{cancer}\;\mathrm{potency}=\left(0.0562\times 0.072\right)+\left(0.0049\times 0.926\right)\;\begin{array}{cc} \mathrm{Average}\;\mathrm{cancer} & \;\mathrm{potency}\\ & =0.0000184\;\mathrm{cases}\;\mathrm{per}\;100,000\;/\mathrm{yr}\;\mathrm{ng}\;\mathrm{aflatoxin}\\ & /\mathrm{kg}\;\mathrm{bw}/\mathrm{day} \end{array}$$

Thus, the population cancer risk of AFM_1_-induced hepatocellular carcinoma (HCC) was determined by multiplying the estimates of mean AFM_1_ exposure with the probability of average cancer potency
(3)$$\mathrm{Cancer}\;\mathrm{risk}\;\left(\mathrm{cases}/100,000\;\mathrm{person}/\mathrm{yr}\right)\;\;\;\;\;\;\;\;\;\;\;\;\;\;\;\;\;\;\;\;\;\;\;\;\;\;\;\;\;\;\;\;\;\;\;\;\;\;\;\;=\mathrm{EDI}\times \mathrm{Average}\;\mathrm{cancer}\;\mathrm{potency}$$

#### 2.7.2. Margin of Exposure

MOEs were calculated by dividing the reference value of 570 ng/kg bw/day (AFM_1_ potency for HCC based on a 2-year study in male Fischer rats) by EDI. MOE value ≥ 10,000 indicates little concern from perspective of a public health issue. BMDL_10_ (benchmark dose level confidence limit of 10%), which is an estimation of the lowest dose that is 95% certain to cause no more than 10% cancer incidence, is recommended for use when calculating MOE. The benchmark dose is the dose that causes a low but measurable response.
(4)$$\mathrm{MOE}=\frac{\mathrm{Reference}\;\mathrm{value}}{\mathrm{EDI}}$$

#### 2.7.3. Hazard Index

The HI was determined by dividing the estimated intake by an acceptable reference value (0.2 ng/kg bw/day) to further assess public health issues through intakes. The TD_50_ of AFM_1_ (the dose that causes tumors in half the tested animals that would have remained tumor-free at zero doses) was divided by an uncertainty factor of 5000 to obtain reference value (0.2 ng/kg bw/day), which is a value equal to a risk level of 1:100,000 [30]. An HI less than 1 generally implies a low risk to consumers of dairy products.
(5)$$\mathrm{HI}=\frac{\mathrm{EDI}}{\mathrm{reference}\;\mathrm{value}}$$

### 2.8. Statistical Analysis

Occurrences of AFM_1_ analysis were carried out in duplicate and the exposure and risk assessment data (Appendix A) were reported as mean and median. One-way analysis of variance (ANOVA) and multiple comparison tests were applied to separate significantly different means across product type (raw milk and cottage cheese) in the study regions. Significance difference was judged at the probability *p* < 0.05. All the analysis was performed using SPSS version 21 software (IBM, Chicago, IL, USA). To summarize the data, Microsoft excel was used. Deterministic health risk assessment model for mean and median of EDI, MOE, HI, and cancer risk were determined using formula.

## 3. Results

### 3.1. AFM_1_ Occurrences in Raw Milk and Cottage Cheese

The concentrations of AFM_1_ in milk products collected from each region are presented in Table 2. In this study, raw milk and cottage cheese were contaminated with different concentrations of AFM_1_. The mean concentrations of AFM_1_ in raw milk and cottage cheese collected from all regions were 0.32 and 0.14 μg/L, respectively. A comparison of milk products showed that raw milk had a significantly (*p* < 0.05) higher concentration of AFM_1_ than cottage cheese. The concentrations of AFM_1_ in raw milk were 2.32-fold higher than in cottage cheese.

In this study, the mean AFM_1_ concentrations of raw milk were between 0.04 and 0.55 µg/L in the study regions. The highest mean AFM_1_ concentration in raw milk was obtained in the Oromia region (0.55 μg/L), followed by SNNP (0.14 μg/L) and the Amhara region (0.04 μg/L). The median values of AFM_1_ concentration were 0.29, 0.10, and 0.03 µg/L for the Oromia, SNNP and Amhara regions, respectively.

In the study regions, the occurrence level of mean AFM_1_ in the cheese samples ranged from 0.08 to 0.24 μg/L. Samples from the Amhara region had a higher level of AFM_1_ (0.24 µg/L), while the SNNP region had a lower level (0.08 µg/L). The median occurrence of AFM_1_ in the Amhara, SNNP, and Oromia regions were 0.15, 0.02, and 0.04 μg/L, respectively.

### 3.2. Estimation of AFM_1_ Exposure

The EDI values of AFM_1_ in the three regions of Ethiopia from the consumption of raw milk and cottage cheese are summarized in Table 2. In the present research, we found mean EDI values for raw milk consumers in the range of 0.0–1.4 ng/kg bw/day in which the highest value was observed in the Oromia region. The mean EDI values recorded for cottage cheese consumers in the Oromia, SNNP, and Amhara regions were 0.14, 0.05, and 0.37 ng/kg bw/day, respectively. In all study regions, raw milk (0.70 ng/kg bw/day) and cottage cheese (0.16 ng/kg bw/day) with an average of 0.38 ng/kg bw/day was obtained.

### 3.3. Risk Characterization and Potential Health Impact in the Study Population

The risk characterization of AFM_1_ through dairy consumption in the study population was characterized using a deterministic approach. The deterministic approach estimates dietary exposure to mycotoxins that might be a health risk for average consumers belonging to certain at-risk population groups [31]. Table 3 shows the values of liver cancer risk, HI, and MOE of AFM_1_ in three regions of Ethiopia from the consumption of dairy products. In the present work, the mean HI values obtained were 3.50 for raw milk and 0.79 for cottage cheese consumers with an average value of 1.90. In all study regions, the mean HI values ranged from 0.0 to 7.05 for consumers of raw milk and local cheese. Both the mean and median values of HI were greater than 1 in the Amhara region for consumers of local cottage cheese.

In this work, the MOE results, 0.0, 1054.6, and 404.2 were recorded in the Amhara, SNNP, and Oromia regions for raw milk consumers, respectively, whereas for cottage cheese consumers, these values were 1534.4, 10,734, and 4111.8 in the above regions respectively. Both the mean and median values of MOE were less than 10,000 in all regions except in the SNNP region where it was greater than 10,000 for cottage cheese consumers.

The cancer risk was estimated based on average potency. The cancer potency was calculated from secondary prevalence data and was 0.0000184 cancers/year/10^5^ persons. The cancer risks due to median exposure to AFM_1_ in the adult populations for raw milk and cheese were 4.3 × 10^−6^ and 7.6 × 10^−8^ cases per year per 10^5^ persons, respectively, whereas the means were 1.29 × 10^−6^ and 2.9 × 10^−6^ cases per year per 10^5^ persons for the above products, respectively. The average cancer risks in the Amhara, SNNP, and Oromia regions were 2.01 × 10^−6^, 4.57 × 10^−6^, and 1.15 × 10^−5^ cases/10^5^/year, respectively.

## 4. Discussions

### 4.1. AFM_1_ Occurrences in Raw Milk and Cottage Cheese

The occurrence of AFM_1_ in raw milk varied in the study regions. The accessibility and use of various amounts of dairy feed that was contaminated due to poor harvest, preharvest handling, and postharvest management (transportation and storage conditions) may be the cause of the variance in AFM_1_ content among regions. Different farming practices, diverse agroecological conditions, and varying climate change may all contribute to raw milk’s variable AFM_1_ levels [32]. AFM_1_ disparity in raw milk could be due to large-volume mixing of various concentrations of AFM_1_-contaminated raw milk from different dairy farmers, through formal and informal means, at the milk collection centers, thereby elevating the concentration of AFM_1_ in mixed raw cow milk. This can, thereby, contribute to the occurrences in its processed products.

AFM_1_ was present in cottage cheese at a lower concentration than it was in raw milk. This could be as a result of the different AFM_1_ contents of the original raw milk produced by different farmers, the mixing of raw milk, and the duration of the fermentation process that results in the destabilization of casein protein during the making of traditional cheese. AFM_1_ occurrence levels are also reduced by fermentation byproducts (lower pH) by creating an acidic environment through the function of lactic acid bacteria during traditional yogurt (*ergo*) preparation. More information is available in previously published work [22]. Quantitative detection of AFM_1_ using the ELISA method is simple, faster, more reliable, and more cost-effective for analyses of large sample sizes. However, it is more sensitive to the detection of aflatoxins in milk products than to their quantification and confirmation compared to other methods (e.g., HPLC) [33].

### 4.2. Exposure Assessment

The risk of dietary exposure to AFM_1_ through dairy consumption in the population was determined using the mean of AFM_1_ concentration, intake of milk products, and body weight. In this work, we found varying values of EDI among regions and dairy products, which could be due to differences in the quantities of milk product intake and AFM_1_ concentrations in the study regions (Table 2). A lower EDI value of raw milk was recorded in the Amhara region than in other regions, which could be due to the comparatively low consumption habit of raw milk intake in the area. In Ethiopia, there is consumption and preference of raw fluid milk due to its flavor, price, and availability among pastoralists in Southern Ethiopia, which may not be representative of the Ethiopian population as a whole [34,35].

In the present investigation, there was a higher value of EDI for local cottage cheese consumers in the Amhara region than in other regions. This can be explained by higher amounts of cheese consumption and higher occurrences of AFM_1_ levels in cottage cheese samples collected from the area. Cottage cheese and other local cheese varieties (*Hazo, Metata*, and *Zureshekefign*) are products that are commonly processed and consumed during the non-fasting season in eastern Gojjam of the Amhara region [36]. The results of the present work only considered exposure and health risk estimations for adults because we had no consumption data for different age categories in the study regions.

The study conducted by [6] reported an estimate of 0.79 ng/kg bw/day mean EDI in adults assuming 70 kg bw, and 0.09 daily consumption of raw milk (kg/day) and AFM_1_ concentration using two previous research findings [15,16], which were slightly higher than the present work (0.70 ng/kg bw/day). They also reported 0.11 EDI in Egypt for raw milk consumers and 0.29–0.59 ng/kg bw/day in Kenya for raw milk, UHT (ultra-treated temperature), and pasteurized milk consumers. The investigation by [37] in southern Ghana obtained between 0.06 and 2.03 ng/kg bw/day EDI in adults (18–64 years) for raw cow milk consumers. All adult consumers in Kenya were exposed to an average of 0.8–1.4 and 0.4–0.7 ng/kg bw/day of AFM_1_ through milk products from low- and middle-income areas, respectively [38]. The mean EDI (0.71 ng/kg bw/day) in raw milk sold in Indian markets was higher for adults [39]. EDI values for Croatian adults obtained from mean raw milk consumption ranged from 0.17 to 2.82 ng/kg bw/day [40]. The results of this work are in agreement with the values mentioned above.

The mean exposure assessments of AFM_1_ for the age group (21–60) for milk and cottage cheese consumers were 1.3 and 0.05 ng/kg bw/day, respectively, in Punjab [41]. The research conducted by [42] found EDI values of 0.16 and 0.20 ng/kg bw/day in adults for traditional soft cheese (*wagashie*) consumers in the Ashanti and Brong Ahafo regions of Ghana, respectively. According to reports from Iran, an adult male and adult female’s lowest EDI value of AFM_1_ from cheese consumption was 0.08 and 0.07 ng/kg bw/day, respectively [43]. AFM_1_ concentrations in milk products from South Asia and sub-Saharan Africa were significantly higher than those allowed by US and EU regulations, suggesting a possible risk to humans for consumers of large quantities of milk products. Populations of children, who might consume more milk than other age groups and may be more susceptible to the negative consequences of exposure to AFM_1_, are of particular concern [6].

### 4.3. Risk Characterization

In this work, higher mean HI values for raw cow milk were observed in the Oromia region, which could be due to higher intake of and higher AFM_1_ concentrations in raw milk (Table 2). In the SNNP and Oromia regions, the mean HI for raw milk was greater than one, indicating a potential health risk in the study populations (both men and women). However, both the mean and median HI values did not cause a public health issue for cottage cheese consumers through exposure in these regions. Conversely, there is a public health issue in the Amhara region for traditional cottage cheese consumers. Generally, the average HI of 1.90 indicates adverse health effects for raw milk and cheese consumers in the adult populations in the study regions (Table 3).

Health risk assessments determined that the HI (3.57) for AFM_1_ in raw milk sold in Indian markets was higher for adults, suggesting a potentially high risk to the consumer’s health [39]. From a study in Croatia, the mean HI for adults via intake of raw cow milk during seasons 2016–2022 (autumn and winter) ranged from 0.88 to 7.62, which implied a high level of heath concern, particularly for consumers of large volumes of milk [40]. In Iran, the HI for cheese consumers who were adult females and males were 0.38 and 0.32, respectively, which indicated no health concern [43]. A similar study by [44] found less than 1 HI values for AFM_1_ levels (using ELISA and HPLC methods) through the consumption of traditional cheeses during the summer and winter. Conversely, a finding by [45] reported that the AFM_1_ level of raw milk in summer had health risks for all age groups of women and men in Iran.

The MOE can be used to characterize the health risk of exposure to carcinogenic and genotoxic substances present in food (milk products) and feed [46]. According to the [7], when the value of the MOE is ≥ 10,000, it is assumed that there is a low risk of a negative influence on public health. Based on this, our results showed that all the mean values of the MOE in all study regions for the mean exposure to AFM_1_ in raw milk and cottage cheese were less than 10,000 in the adult populations, which indicates a potential public health issue due to AFM_1_ exposure through intake (Table 3). However, it was not a concern for consumers of cottage cheese in the SNNP region. In Ghana, MOE values (197–6666.7) were found in adults for raw cow milk consumers, which implies a public health issue [37].

The fraction of liver cancer or HCC incidence attributed to AFM_1_ consumption was taken into account based on the MOE using the estimated mean exposure. [46]. In the study regions, the cancer risk due to mean AFM_1_ exposure in the adult populations for raw milk and cheese was lower. In line with this study, it was observed that in Kenya, the annual risk for HCC per 100,000 persons for lower income consumers for all milk categories was 1.6 × 10^−5^–2 × 10^−2^ and 7.0 × 10^−5^–7 × 10^−3^, respectively. This indicates a low risk for cancer due to AFM_1_ exposure from milk consumption for adults [38].

In a study in Malawi, the risk of AFM_1_-induced HCC associated with the intake of milk in the adult population was 0.023 cases/year/10^5^, suggesting a low risk of HCC [47]. The risk of HCC cases per year per 10,000 individuals in Punjab who were of various ages revealed that the value of HCC using a deterministic approach was (0.002) in the age group “21–60 years” [41]. Based on the mean consumption and yearly weighted mean AFM_1_ concentration, the estimated fractions of HCC incidence cases in Italy were between 0.0004 and 0.0008 per 100,000 people for adults [46]. The fraction of HCC cases/100,000 individuals for milk products consumed by adults (bw, 60 kg) of north Macedonia was 0.004 [48]. The results of this work and the above findings are in agreement with [6] who reported no synergy between AFM_1_ and HBV, which may contribute 0.001% of total annual HCC cases globally. However, with synergy between AFM_1_ and HBV infection, AFM_1_ may contribute approximately 0.003% of all HCC cases worldwide. However, In Ghana, 3.5 × 10^−3^–0.06 cases/10^5^ person/year were recorded for local cheese [42], which posed potential health effects on all age groups in the study areas.

The most aflatoxin-related health problems are linked to the primary liver cancer burden since consuming these toxins has been found to significantly increase the risk of developing HCC, particularly in people who already have hepatitis virus infection. There is still no uncertainty on the consequences of AFM_1_ on health, particularly the combined effects of combinations of mycotoxins, aflatoxins, and others food sources on cancer risk [38].

## 5. Conclusions

In conclusion, all samples collected from all regions were detected with different concentrations of AFM_1_. According to the deterministic approach parameters of MOE and HI values, the health risk results in the current study suggested that the mean AFM_1_ exposure through the consumption of raw milk and cottage cheese indicates potential public health impacts. However, there is a low risk for cancer due to exposure to AFM_1_ from raw milk and cottage cheese intake in adult populations. Strict regulation and monitoring of animal feeds in the feed value chain is crucial in minimizing AFM_1_ contamination of dairy products to the acceptable level. Moreover, a health risk assessment of AFM_1_ in children should be studied further as their consumption is higher.

## Figures and Tables

**Table 1 foods-12-00817-t001:** Composite sample size of milk products along the value chain in Ethiopia.

Value Chain	Product Type	Study Regions			Total
		Oromia	SNNP	Amhara	
Farmers	Raw milk	16	8	8	32
	Cottage cheese	8	4	4	16
Collectors	Raw milk	16	8	8	32
Retailer	Cottage cheese	8	4	4	16
Total		48	24	24	96

**Table 2 foods-12-00817-t002:** Exposure estimation of AFM_1_ in adult populations in three regions of Ethiopia.

Region	Product Type	Ave. Intake (L or kg)	Ave. Bw (kg)	AFM_1_ Concentration (μg/L)		EDI (ng/kg bw/day)	
				Mean	Median	Mean	Median
Oromia	Raw milk	0.15	57.58	0.55 ^a^	0.29	1.41	0.75
	Cottage cheese	0.07	57.58	0.12 ^b^	0.04	0.14	0.05
Ave.		0.11	57.58	0.33	0.17	0.62	0.31
SNNP	Raw milk	0.23	57.42	0.14 ^a^	0.10	0.54	0.40
	Cottage cheese	0.04	57.42	0.08 ^b^	0.02	0.05	0.02
Ave.		0.13	57.42	0.11	0.06	0.25	0.14
Amhara	Raw milk	0.00	57.04	0.04 ^a^	0.03	0.00	0.00
	Cottage cheese	0.09	57.04	0.24 ^b^	0.15	0.37	0.24
Ave.		0.04	57.04	0.14	0.09	0.11	0.07
Product Type	Raw milk	0.13	57.34	0.32 ^a^	0.08	0.70	0.34
	Cottage cheese	0.07	57.34	0.14 ^b^	0.04	0.16	0.03
Ave.		0.10	57.34	0.23	0.06	0.38	0.14

AFM_1_ values were duplicate analysis (n = 2); mean values in column with small letter superscript are significantly different (*p* < 0.05); EDI = estimated daily intake; LOD = 0.002 µg/L.

**Table 3 foods-12-00817-t003:** Risk characterization of AFM_1_ through intake of raw milk and cottage cheese in the three regions of Ethiopia.

Study Region	Product Type	HI		MOE		Cancer risk (Cases/10^5^ Person/Year)	
		Mean	Median	Mean	Median	Mean	Median
Oromia	Raw milk	7.05 ^a^	3.76	404.2 ^a^	757.4	2.59 × 10^−5 a^	1.38 × 10^−5^
	Cottage cheese	0.69 ^b^	0.23	4111.8 ^b^	12,335.3	2.55 × 10^−6 b^	8.50 × 10^−7^
Ave.		3.12	1.56	912.1	1829.6	1.15 × 10^−5^	5.73 × 10^−6^
SNNP	Raw milk	2.70 ^a^	1.98	1054.6 ^a^	1436.1	9.95 × 10^−6 a^	7.3 × 10^−6^
	Cottage cheese	0.27 ^b^	0.08	10,734 ^b^	34,438.2	9.77 × 10^−7 b^	3.05 × 10^−7^
Ave.		1.24	0.72	2293.3	3946.0	4.57 × 10^−6^	2.66 × 10^−6^
Amhara	Raw milk	0 ^a^	0	0 ^a^	0	0 ^a^	0
	Cottage cheese	1.86 ^b^	1.19	1534.4 ^b^	2398.1	6.84 × 10^−6 a^	4.37 × 10^−6^
Ave.		0.55	0.35	5210.2	8183.3	2.01 × 10^−6^	1.28 × 10^−6^
Product Type	Raw milk	3.50 ^a^	1.68	815.4 ^a^	1698.8	1.29 × 10^−6 a^	4.30 × 10^−6^
	Cottage cheese	0.79 ^b^	0.13	3621.8 ^b^	21,720.1	2.9 × 10^−6 b^	7.60 × 10^−8^
Ave.		1.90	0.71	1497.0	3988.9	7.0 × 10^−6^	1.32 × 10^−6^

Mean values in column with small letter superscript are significantly different (*p* < 0.05); MOE = margin of exposure; HI = hazard index.

## Data Availability

Data is contained within the article or Appendix A.

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
