# Peer review of "Exposure and Health Risk Assessment of Aflatoxin M1 in Raw Milk and Cottage Cheese in Adults in Ethiopia"

_foods, 2023, doi:10.3390/foods12040817_

Round 1
Reviewer 1 Report
Dear authors,
Comments are given in the attached file.

Author Response
Rebuttal letter
Manuscript ID: foods-2094239
Title: Exposure and health risk assessment of Aflatoxin M1 in raw milk and cottage cheese in adults in Ethiopia
We appreciate both the editor and reviewers for your valuable comments and time for the improvement of the manuscript. Please find below our responses to the reviewer comments. We have also revised the manuscript accordingly and highlighted the trach changes in red color inside the manuscript.
Authors’ responses to the reviewer comments
We made changes the subtitle ‘sample collection’, with sample size and sampling. Please see line 136.
We made changes the ELISA method sentence. Please see line 155-156.
We excluded pasteurized milk from the method (Please see line 200-201). We also made changes in the sample size of composite samples in Table 1.
We have also moved the limitation of the study into discussion section. Please see line 449-451.

Reviewer 2 Report
Very important information well put across to the readers of this manuscript to better improve their public health safety

Author Response
Rebuttal letter
Manuscript ID: foods-2094239
Title: Exposure and health risk assessment of Aflatoxin M1 in raw milk and cottage cheese in adults in Ethiopia
We appreciate both the editor and reviewers for your valuable comments and time for the improvement of the manuscript. Please find below our responses to the reviewer comments. We have also revised the manuscript accordingly and highlighted the trach changes in red color inside the manuscript.
Authors’ responses to the reviewer comments
Comments and Suggestions for Authors
Very important information well put across to the readers of this manuscript to better improve their public health safety.
Authors’ response: Thank you very much for your message.
- Line 37- The results indicted …….
Authors’ response: We made changes accordingly. Please see line 39.
- Line 40- Our results showed that……
Authors’ response: We made changes accordingly. Please see line 42.
- Authors should please label all the equations stated in the manuscript for easier identification
Authors’ response: We labeled all the questions in the manuscript accordingly.
- Please state the kind of risk assessment method used i.e deterministic or probabilistic
Authors’ response: We stated this accordingly. Please see line 361-363.
- The assumption of 65 kg used must have a reference
Authors’ response: We have stated the references for the average body weight from recently two sources that they used for exposure estimations (Please see line 200). There is still limited data on average bw of men adults in Ethiopia.
- Authors must state the age range that makes an individual an adult in Ethiopia
Authors’ response: We have stated the age range that makes an individual an adult in Ethiopia. Please see line. 195-197.
- Authors should have considered the different age categories in the population since it gives a
better idea of the state of intoxication.
Secondly. Adults are not the only consumers of these milk products
Authors’ response: We have put this as limitation of the study and further research work to be done (Please see line 449-451). Because there is no dairy products consumption data at different age categories in the study regions.
- There should be a separate portion in the manuscript for the statement of results and discussion
for the quantities/levels of the AFM1 in the milk products
Authors’ response: We put a separate portion statement for occurrences of the AFM1 in the milk products in results (please see line 306-325) and discussion (please see line 423-429 and 444-451).
- In the discussion, possible causes of the observed variations must be stressed on.
Authors’ response: We have added possible causes of the observed variations in the discussion section. Please see line 426-429 and 445-447.
- What were the LOD and LOQ of the AFM1 using ELISA method.
Authors’ response: We have included LOD for the method. Please see line 173-175.
Reviewer 3 Report
abstract: erase the phrase “However, AFM1 contamination of milk poses a health risk” because this is the matter of your objective
keywords: replace raw milk with dairy (keywords must be different from the title)
Revise English, form and redaction in all manuscript. For example: L150-152, L 93-94 “So far only three were published [15,16] have been published”. Why are there 2 references for 3 studies?
L 99 (and others) Name the institution or name, then add the reference.
not put the DOI on the manuscript, add a reference and explain briefly
In methods, not use ppb, replace it for μg/L (all manuscript). For the low numbers it is convenient to use ng/L
L160: just put the reference number for methodology. Summarize the validation of the method to explain the LODs and LOQs of the method
For deterministic models, is useful to estimate a worst case scenario, with a P95 of levels and high consumption
formulas of 2.8.1 if you put a value, you have to add the unit
statistical analysis: add occurrences differences between processings, places and types of dairy analyzed. ELISA is more sensitive for occurrence than in levels
Where are the results of AFM1 for pasteurized milk? If is presented in methods, must be in results
In table 2 add LOD and LOQ of the method in the footer
Table 3 abbreviations in footer. Add statistical differences between sources, region..
Add limitations of the study in the discussion section
Will be useful the AFM1 data in the different sources as a supplementary material
Author Response
Rebuttal letter
Manuscript ID: foods-2094239
Title: Exposure and health risk assessment of Aflatoxin M1 in raw milk and cottage cheese in adults in Ethiopia
We appreciate both the editor and reviewers for your valuable comments and time for the improvement of the manuscript. Please find below our responses to the reviewer comments. We have also revised the manuscript accordingly and highlighted the trach changes in red color inside the manuscript.
Authors’ responses to the reviewer comments
Comments and Suggestions for Authors
abstract: erase the phrase “However, AFM1 contamination of milk poses a health risk” because this is the matter of your objective.
Authors’ response: We deleted it accordingly. Please see line 34-35.
keywords: replace raw milk with dairy (keywords must be different from the title).
Authors’ response: Thank you. We replaced raw milk with dairy accordingly. Please see line 50.
Revise English, form and redaction in all manuscript. For example: L150-152, L 93-94 “So far only three were published [15,16] have been published”. Why are there 2 references for 3 studies?
Authors’ response: We have revised the English in the line 155-156 and line 96-97, and in the manuscript too. We also include the missed one references. Please see line 97.
L 99 (and others) Name the institution or name, then add the reference.
Authors’ response: We made changes the citations in the line 103 and throughout the manuscript accordingly.
not put the DOI on the manuscript, add a reference and explain briefly.
Authors’ response: We replaced DOI of the manuscript by adding the references and we explained it briefly accordingly. Please see line 176 and line 429.
In methods, not use ppb, replace it for μg/L (all manuscript). For the low numbers it is convenient to use ng/L.
Authors’ response: We made changes the units throughout the manuscript accordingly. Please see line 162 and 174.
L160: just put the reference number for methodology. Summarize the validation of the method to explain the LODs and LOQs of the method.
Authors’ response: We put our reference number (please see line 174) and we also summarized the method of validation accordingly (please see line 169-175).
For deterministic models, is useful to estimate a worst case scenario, with a P95 of levels and high consumption.
Authors’ response: It would be good through probabilistic modeling in the future works to provide better information, with additional comprehensive consumption data in different age categories.
formulas of 2.8.1 if you put a value, you have to add the unit.
Authors’ response: We put the unit accordingly. Please see line 251.
Statistical analysis: add occurrences differences between processings, places and types of dairy analyzed. ELISA is more sensitive for occurrence than in levels.
Authors’ response: We have added this in the statistical analysis subsection accordingly. Please see line 284-287.
Where are the results of AFM1 for pasteurized milk? If is presented in methods, must be in results.
Authors’ response: Yes. We agree with you. We excluded from methods (Please see line 200-201). We also made changes in sample size of composite samples in Table 1. Because, there is additional information in previously published work.
In table 2 add LOD and LOQ of the method in the footer
Authors’ response: We added LOD of the method in Table 2 footer accordingly. Please see line 345.
Table 3 abbreviations in footer. Add statistical differences between sources, region.
Authors’ response: We put abbreviation in the footer (Table 3) accordingly. We also added the statistical differences between sources in the regions.
Add limitations of the study in the discussion section.
Authors’ response: We have added the limitation of the study accordingly. Please see line 449-451.
Will be useful the AFM1 data in the different sources as a supplementary material.
Authors’ response: Yes, we agree with you. We included it in the supplementary material. Please see line 557.

Round 2
Reviewer 3 Report
Abstract: connect the first sentence with the second or erase the first sentence. You can start explaining what is AFM1
Which were the recoveries and variation coefficients of the method validation? Inform the concentrations of the spike samples and the recoveries (at least 3 points)
Which was the range of detection and quantification of the method? If you quantify, you MUST put a LOQ value. And, you found 550 ng/L, you have to demonstrate that the method can quantificate at this level.
Did you use the kit´s LOD or the best recovery (you put 3 different LODs)? Did you use the same LOD and LOQ value?
If you put the LOD in ng/L, the results have to be in that unit; or put the LOD and LOQ in ug/L
Formulas 4 and 5: put a unit in the numerical values or erase them
The limitations of the method are still missing.
Author Response
Rebuttal letter
Manuscript ID: foods-2094239
Title: Exposure and health risk assessment of Aflatoxin M1 in raw milk and cottage cheese in adults in Ethiopia
We appreciate again both the academic editor and reviewers for your valuable comments and time for the improvement of the manuscript to better quality. Please find below our responses to the reviewer comments. We have also revised the manuscript accordingly and highlighted the trach changes in red color inside the manuscript.
Authors’ responses to the reviewer comments (second round)
Comments and Suggestions for Authors by the Academic Editor Notes
Please revise your manuscript according the latest comments of the reviewers:
- Abstract: connect the first sentence with the second or erase the first sentence. You can start explaining what is AFM1
Authors’ response: We have made changes in the abstract start by explaining what is AFM1 accordingly. Please line 34-35.
- Which were the recoveries and variation coefficients of the method validation? Inform the concentrations of the spike samples and the recoveries (at least 3 points).
Authors’ response: We put the recoveries (please see line 172-174) and CV values of the method validation (Please see line 175-176 for intra-assay precision and line 177-178 for inter-assay precision). We have also informed the lowest concentration samples used for spiking. Please see line 174-175.
- Which was the range of detection and quantification of the method? If you quantify, you MUST put a LOQ value. And, you found 550 ng/L, you have to demonstrate that the method can quantificate at this level.
Authors’ response: The range of detection and quantification of the method is between 0.005 to 0.1 µg/L. Please see herein, the next page the raw data for calculating recovery and reproducibility that we did during the study period and
in the supplementary material (https://www.mdpi.com/article/10.3390/toxins14040276/s1, Spreadsheets S1: ELISA raw data set.)
- Did you use the kit´s LOD or the best recovery (you put 3 different LODs)? Did you use the same LOD and LOQ value?
Authors’ response: We used kits LOD 0.002 µg/L. We deleted for raw milk and cheese LOD (please line 179-180).
- If you put the LOD in ng/L, the results have to be in that unit; or put the LOD and LOQ in ug/L
Authors’ response: We only put LOD in µg/L same to results.
- Formulas 4 and 5: put a unit in the numerical values or erase them
Authors’ response: We have erased the numerical values accordingly. Please see formula 4 and 5.
- The limitations of the method are still missing.
Authors’ response: We have mentioned limitation of the method in the discussion section. Please see line 435-438.
